

# A Spectral Perspective of ENSO Driven OLR Variability

Martina Taddia[1,2], Federico Fabiano[2], Stefano Della Fera[3], Elisa Castelli[2], and Bianca Maria Dinelli[2]

[1]Department of Physics and Astronomy, University of Bologna, via Irnerio 46, 40126, Bologna, Italy
[2]Institute of Atmospheric Sciences and Climate, National Research Council (ISAC-CNR), Via Piero Gobetti, 101, 40129 Bologna, Italy
[3]Institute of Applied Physics, National Research Council (IFAC-CNR), Sesto Fiorentino (FI), Via Madonna del Piano, 10, 50019 Sesto Fiorentino, Firenze, Italy

**Correspondence:** Martina Taddia (martina.taddia8@unibo.it)

**Abstract.** The study of short-term unforced variability of the Earth radiative budget can provide much information for the understanding of the long-term effect of external radiative forcing, related to the present climate change. In this regard, inter-annual variability of the Outgoing Longwave Radiation (OLR) is strongly shaped by El-Niño Southern Oscillation (ENSO). So far, the relationship between the OLR and ENSO has been investigated using broadband satellites-based observations, such as those of the Clouds and Earth Radiant Energy System (CERES), finding that the peak of the OLR response lags the peak of ENSO activity. However, such analysis cannot inform on the individual processes that drive the radiative response to ENSO. Here, we exploit the spectrally-resolved clear-sky OLR fluxes - measured by the Infrared Atmospheric Sounding Interferometer (IASI) and the Atmospheric Infrared Sounder (AIRS) instruments - to expand the observational analysis of ENSO's radiative response. The observed signal is then decomposed using a spectral kernel analysis into water vapor, surface and air temperature, and ozone feedback, to evaluate the role of individual processes building the overall response. Results show a strong contribution coming from the ozone absorption band, along with a contribution of opposite sign coming from the the core of the carbon dioxide band, which is mainly affected by stratospheric temperature. This analysis confirms the important role of the spectral dimension to study climate processes. In this regard, it sets the basis for a spectral diagnostic to evaluate how ENSO driven variability is reproduced by climate models.

## 1 Introduction

The Outgoing Longwave Radiation (OLR) is the energy flux emitted by the Earth towards space across the thermal and far infrared (FIR) spectral regions (from 100 to about 3330 $cm^{-1}$, 3-100 $\mu m$) and represents the mechanism by which the climate system loses energy to maintain its equilibrium. It contains both the signature of the increased concentration of radiatively active species - such as carbon dioxide and other Greenhouse Gases (GHGs), driving present-day climate change - and that of the system's radiative response, produced by the warming of planet's surface and resulting climate feedbacks. The exact magnitude of climate feedbacks, which determine the ultimate warming of the planet in response to forcing, is still uncertain and global climate models (GCMs) struggle to agree on a precise value. For example, a stronger positive clouds feedback in Coupled Models Intercomparison Project phase 6 (CMIP6) is the cause of a higher Effective Climate Sensitivity (ECS) estimate, with



values between 1.8 - 5.6 K, exceeding the range 2.1 - 4.7 K, obtained with the previous generation (CMIP5) (Zelinka et al.,
2020). On the other hand, several satellite instruments have been deployed to measure the OLR from space, but the available
observational record is not yet long enough to accurately quantify the long-term trend and use it to constraint climate models
feedbacks (Uribe et al., 2024). A possible way ahead is represented by the study of the planet's internal variability. The OLR is
in fact also subjected to fluctuations on seasonal and interannual timescales and several research efforts have been focused on
the study of the short-term variability to improve the understanding of feedback processes on longer timescales (Dessler et al.,
2008; Dessler, 2010, 2013; Uribe et al., 2022).

On the inter-annual time scale, El-Niño Southern Oscillation (ENSO) is the dominant variability mode of the climate system.
It causes changes in the Tropical Pacific Sea Surface Temperature (SST) coupled with changes in the above atmospheric
circulation, affecting the Earth climate on a global scale (McPhaden et al., 2020). ENSO induced perturbations to atmospheric
and surface variables strongly affect the Top-Of-Atmosphere (TOA) OLR. As shown by Loeb et al. (2012) and Susskind et al.
(2012), the effect of ENSO is detectable in the OLR measured by the Clouds and Earth Radiant Energy System (CERES), which
co-varies significantly with ENSO activity, assuming positive (negative) anomalies in correspondence to El-Niño (La-Niña)
phase. The main radiative feedback driven by ENSO induced SST perturbations involves water vapor, temperature and clouds
properties changes (Huang et al., 2021). Their evolution during the ENSO life cycle provides information on the feedbacks
important for ENSO developing and maintenance. Particularly, the atmosphere exerts an overall positive feedback that act to
strengthen ENSO during its developing phase (Kolly and Huang, 2018; Huang et al., 2021).This result is also confirmed by
Ceppi and Fueglistaler (2021), who use broadband measurements to investigate the time relationship between the net TOA
Earth Radiative Budget and ENSO. Ceppi and Fueglistaler (2021) show that the SST pattern effect (Stevens et al., 2016) is
responsible for the out-of-phase relationship between the radiative response and SST anomalies, with the first leading the
ENSO peak, thus suggesting a two-way coupling between the atmosphere and ocean.

The study of the radiative response to ENSO potentially allows to better constrain climate feedbacks and evaluate climate
model performance, inspecting their representation of the coupled atmosphere-ocean dynamics (Andrews et al., 2015; Armour
et al., 2024; Andrews et al., 2022). Indeed, there are evidences that the radiative response to ENSO is still not completely
reproduced in both its amplitude and timing (Kolly and Huang, 2018; Planton et al., 2021; Ceppi and Fueglistaler, 2021).
While the SST pattern effect plays an important role for the relationship between unforced and forced feedbacks that can be
used as an emergent constraint for climate models (Davis et al., 2024; Rugenstein et al., 2025). Hanke and Proistosescu (2024)
propose ENSO as an emergent constraint on the pattern effect.

The present work aims to demonstrate the value of the spectral dimension in OLR measurements for the understanding of
the mechanisms that drive the radiative response to ENSO. So far, OLR variability driven by ENSO has been examined using
broadband OLR, which provide the total radiance integrated over the IR spectral range (e.g. 50 - 2000 cm$^{-1}$ for CERES).
The main advantage of the spectral dimension is the possibility to isolate single contributions by examining different spectral
regions and to identify the atmospheric layers where the radiation originates (Brindley and Bantges, 2016). The trend of the first
ten years of observations acquired by the Infrared Atmospheric Sounding Interferometer (IASI), shows the spectral signature
related to both the increase in GHGs concentration, such as carbon dioxide and methane, and the signal related to ENSO





variability across channels sensitive to water vapor and temperature (Whitburn et al., 2021). Using IASI observations, Roemer
et al. (2023) calculated the LW spectral feedback parameter and directly observed the spectral fingerprint due to changes in
relative humidity for an increasing value of global mean surface temperature. Moreover, spectral OLR have been proven to be
a powerful method to assess climate model performances allowing to reveal biases otherwise hidden by compensation effects
when broadband fluxes are considered (Della Fera et al., 2023; Huang et al., 2007; Leroy et al., 2008; Huang et al., 2014).

Building on the analysis by Ceppi and Fueglistaler (2021), we first assess the OLR response to ENSO using CERES broad-
band observations. We then explore the potentiality of the spectral dimension to identify which processes drive the timing and
the magnitude of the OLR anomalies, comparing measurements acquired by two hyperspectral sounders, the Infrared Atmo-
spheric Sounding Interferometer (IASI) and the Atmospheric Infrared Sounder (AIRS). Finally, the role of water vapor, surface
and air temperature spectral feedback is investigated, calculating their radiative changes by means of spectral radiative kernels.
We develop a new diagnostics that highlights the role of different feedback processes, and propose to apply it for a stricter
evaluation of climate model performance.

The article is organized as follows: a description of the datasets and of the methodology applied is provided in section 2; the
results of the analysis are presented in section 3; the main findings are discussed in section 4, and final conclusions are drown
in section 5.

## 2 Data and Methods

The analysis uses monthly mean OLR fluxes in clear-sky conditions derived from CERES, AIRS and IASI observations.

CERES is a broadband radiometer (Wielicki et al., 1996) with three channels that measure radiances across the whole
spectrum (0.3-200 $\mu$m), the shortwave (SW, 0.3-5 $\mu$m) and the atmospheric window (8-12 $\mu$m) regions respectively. We use
the CERES energy balanced and filled (EBAF) edition 4.1 data product (Loeb et al., 2020), which gathers data collected by
different CERES instruments on board Terra and Aqua platforms since 2000 (data collection is still ongoing). Two clear-sky
products are available, one that only considers cloud-free observations and another that additionally applies a correction factor
for a better comparison with climate models output, for which clear-sky fluxes are artificially produced by removing clouds
from the flux calculation. For this work the first product has been selected, for consistency with IASI and AIRS observations.
CERES-EBAF 4.1 provides monthly mean TOA OLR fluxes with an horizontal resolution of 1° x 1° and a temporal coverage
from January 2000 to March 2022.

AIRS is a grating array spectrometer that has flown on board Aqua satellite since 2002 (Aumann et al., 2003). It covers three
different bands, 650-1136 cm$^{-1}$, 1216-1613 cm$^{-1}$, 2170-2674 cm$^{-1}$. Monthly mean OLR spectral fluxes have been derived
by Huang et al. (2008, 2019) for both clear and all-sky conditions. AIRS fluxes are calculated exploiting the same angular
distribution models of CERES, for this reason the computation has been performed only for AIRS pixel that overlap up to
a certain threshold with CERES ones. In the clear-sky scenario only clear-sky CERES observations are considered affecting
the final AIRS spatial sampling. The original spectral range has been filled with simulations and extended to the FIR spectral



**Table 1.** Main features of the datasets used.

|  | CERES | AIRS | IASI |
|---|---|---|---|
| Temporal Resolution | monthly | monthly | monthly |
| Temporal Coverage | 01/2000-12/2020 | 09/2002-12/2021 | 01/2008-10/2021 |
| Horizontal Resolution | 1°x1° | 2°x2° | 2°x2° |
| Spectral Coverage ($cm^{-1}$) | 50-2000 | 15-1995 | 645-2300 |
| Reference | *Loeb et al. (2020)* | *Huang et al. (2008)* | *Whitburn et al. (2020)* |

region (100-600 $cm^{-1}$), so that the final product consists of monthly mean spectral fluxes computed every 10 $cm^{-1}$ from 15 to 1995 $cm^{-1}$. AIRS data range from November 2008 to December 2021, with a horizontal resolution of 2° x 2°.

IASI is a Michelson interferometer which measures radiances across the 645-2760 $cm^{-1}$ spectral range with a resolution of 0.25 $cm^{-1}$ (Simeoni et al., 2004). It has flown on board MetOp-A (10/2007-12/2021) satellite and it is currently operating on
board the MetOp-B (2013-) and C (2013-). We use the monthly climatology developed by Whitburn et al. (2020); Whitburn (2021b) using observations collected by IASI instrument on board the MetOp-A, which has the longest temporal coverage, from January 2008 to October 2021. Spectral fluxes are provided from 645 up to 2300 $cm^{-1}$ (to avoid contamination of shortwave radiances) keeping the original resolution of 0.25 $cm^{-1}$ and for only clear-sky scenario. They have an horizontal resolution of 2° per 2° latitude. The main features of the datasets are resumed in Table 1.

All the datasets have been regridded to a common grid of 2.5° latitude per 2.5° longitude, through a bilinear interpolation. We focused on the tropical zone (30°S-30°N), where the radiative perturbations induced by ENSO are strongest, and on values above ocean, to avoid introducing biases in the comparison of non-simultaneous measurements, which can arise from the rapid temperature variability typical of land surfaces (Huang and Yung, 2005; Whitburn et al., 2021). The period from 01/2008 to 12/2020, common to all the observational datasets, has been selected for the analysis. We computed monthly anomalies
removing the seasonal cycle and the linear trend for the whole period.

As a measure of the ENSO phase over the considered period, we use the Niño 3.4 index. Following Trenberth (1997), deseasonalized and detrended SST anomalies have been calculated for the Niño 3.4 region (5°S-5°N,170°-120°W), and then smoothed with a 5-months running mean. Area averaged SST are provided by the NOAA-PSL database (https://psl.noaa.gov/data/correlation/nina34.anom.data) to calculate the Niño 3.4 index.

We use lagged regressions to investigate the time relationship between the OLR and ENSO. Specifically, radiative anomalies have been regressed onto the Niño 3.4 index shifted one month at a time from -12 to +12 month (25 slopes are obtained at the end of the procedure). The 95% statistical significance of the slopes has been assessed with a two-tailed t-test. The regression coefficients are a measure of the radiative perturbations driven by ENSO induced SST anomalies, to which we will refer to as "ENSO feedback" during the rest of the paper, as done also by Kolly and Huang (2018); Huang et al. (2021).





## 2.1 Spectral Radiative Kernels

To disentangle the contributions of atmospheric temperature, water vapor, ozone and surface temperature to the change in the radiative fluxes driven by ENSO, we apply a spectral kernel analysis. Radiative kernels (or Jacobians) represent the partial derivative of spectral flux with respect to specific climate variables (temperature, surface temperature, water vapor and ozone), evaluated across the atmospheric levels in clear-sky conditions. These derivatives are computed using the Radiative Transfer for TOVS (RTTOV v13) model (Matricardi, 2009), based on monthly mean profiles from the ECMWF Reanalysis v5 (ERA5) dataset for the years 2008, 2009, and 2010, and then averaged to obtain representative monthly kernels. As done by Huang et al. (2014), at least one year of data is sufficient for kernels computation.

The RTTOV model is customized to reproduce the synthetic radiance of specific instruments working in different spectral ranges, from visible to microwaves. In our case, since we aim to compute the derivative across the entire Earth infrared spectral range from 100 to 2300 $cm^{-1}$, we calculated the derivative of synthetic radiance for FORUM (the EE-9 selected by ESA, Far-Infrared Outgoing Radiation Understanding and Monitoring) in the FIR spectral region (100–650 $cm^{-1}$), using its associated instrumental function, as well as that of IASI in the MIR spectral range (650–2300 cm$^{-1}$). Furthermore, in order to obtain the derivatives of the spectral flux (W m$^{-2}$ (cm$^{-1}$)$^{-1}$), radiance derivatives (W $m^{-2}(cm^{-1})^{-1}$ sr$^{-1}$) were computed at three distinct viewing angles (24.29°, 53.80°, 77.74°). The resulting values were then integrated using Gaussian quadrature, which has been shown to yield accurate results for clear-sky radiative transfer calculations (Clough et al., 1992). The resulting dataset consists of radiative kernels, averaged over 10 cm$^{-1}$ intervals, from 100 to 2300 cm$^{-1}$, spanning 17 pressure levels (hPa): 1000, 925, 850, 700, 600, 500, 400, 300, 250, 200, 150, 100, 70, 50, 30, 20, 10, on a horizontal grid of 2.5° × 2.5°.

The perturbation to the radiative flux is obtained multiplying the kernel by the anomaly of the variable of interest and then, for 3D variables, summing over the pressure levels to obtain the total atmospheric response.

Atmospheric variables are taken from the monthly ERA5 reanalysis (Hersbach et al., 2023b), selected on the same 17 pressure levels used for the radiative kernels. We consider the period 2008–2020, and compute the anomalies relative to the mean over the entire time span.

## 3 Results

We present here the main results of the analysis, starting from the broadband fluxes and then moving to the spectral dimension.

## 3.1 Broadband radiative response to ENSO

The slopes of the lagged regressions between the OLR anomalies and the Niño 3.4 index performed for broadband fluxes are reported in Figure 1 as a function of lags. The sign of the OLR is defined as positive upward, meaning that for positive slopes an increase in OLR emission takes place, and vice versa for negative. For both AIRS and IASI, the radiative fluxes have been integrated along the spectral dimension. In the case of AIRS, two broadband fluxes have been calculated: one using spectral channels from 50-1995 cm$^{-1}$ (dashed line), to be as consistent as possible with CERES (encompassed range from 50 to 2000




cm$^{-1}$), and a second limited to the IASI spectral range (645-1995 cm$^{-1}$). For both the two spectral range encompassed, the magnitude of the slopes increases moving from negative to positive lags and reaches a maximum at a lag of 2 months, showing that ENSO is associated to an increase in the TOA OLR emission. When both the MIR (650–2300 cm$^{-1}$) and the FIR spectral region (100–650 cm$^{-1}$) are accounted for, the maximum slope is of $0.46 \pm 0.11$ Wm$^{-2}$K$^{-1}$ for AIRS and $0.46 \pm 0.12$

150    Wm$^{-2}$K$^{-1}$ for CERES, while its value is of $0.31 \pm 0.05$ Wm$^{-2}$K$^{-1}$ for AIRS and $0.31 \pm 0.05$ Wm$^{-2}$K$^{-1}$ for IASI, when only the MIR is considered. This denotes that an important part of OLR emission during ENSO is achieved across the FIR, confirming its importance for the TOA Earth Radiative Budget (ERB). Although we find an excellent agreement in the value of the maximum slope between CERES and AIRS, there is a slight delay in the AIRS signal, with the peak delayed by 1 month (at 3-month lag) and a smaller slope at negative lags. The difference is likely due to AIRS's clear-sky scene selection, as will

155    be investigated further in Section 4. The agreement between AIRS and IASI is excellent, with minor differences at negative lags.

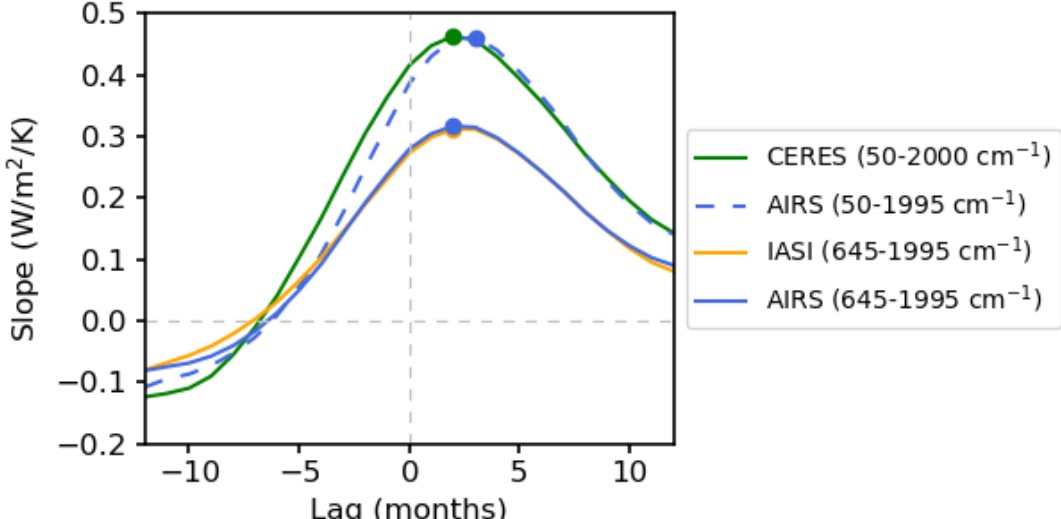

**Figure 1.** Lagged regressions of the tropical mean integrated OLR anomalies from CERES (green), AIRS (blue) and IASI (orange), respect to the Niño 3.4 index, from lag -12 to +12 months.

## 3.2    ENSO spectral footprint from AIRS and IASI

We now wish to examine the factors controlling the magnitude and timing of the response peak. To do so, the analysis has been extended to the spectral dimension and lagged regressions have been performed for each spectral channel of IASI and AIRS.

160    To allow for a direct comparison, IASI has been resampled at the same resolution of AIRS (10 cm$^{-1}$).

The maps in the central and bottom panels of Figure 2 show the regression slopes (unit of Wm$^{-2}$K$^{-1}$) obtained as a function of wavenumber and lag for IASI and AIRS. The top panel shows the slopes as a function of wavenumber alone for 2-month lag, where the broadband regression slopes peak.



The first thing to note is that a different response is obtained at different wavenumbers, which are associated to a characteristic lag, sign and magnitude. A positive peak is found between 1000 - 1065 cm$^{-1}$, at 3-month lag, where the ozone absorption band, centered at 1043 cm$^{-1}$, is located. This constitutes the response with the highest magnitude across the whole spectrum. Other important contributions come from the wings of the carbon dioxide absorption band ($\sim$ 600-645 cm$^{-1}$ and $\sim$ 700-800 cm$^{-1}$), associated to positive slopes with a lag of 3-month, and from its center ($\sim$ 645-700 cm$^{-1}$), characterized by negative slopes at 5-month lag. The two atmospheric windows (800-1000 cm$^{-1}$ and 1100-1200 cm$^{-1}$) show both positive slopes centered at lag equal to +2 months. Below 600 cm$^{-1}$ we can relay only on AIRS simulated spectral fluxes. These wavenumbers exhibit positive slopes, too, of higher magnitude between $\sim$ 400-600 cm$^{-1}$, than between $\sim$ 100-400 cm$^{-1}$. These two spectral regions also differ in the time of the response, centered at 3-month lag for the first and at 2-month lag for the second.

Spectral footprints for AIRS and IASI are generally in a good agreement, showing the same main features. Nevertheless, there are some differences in the magnitude of the slopes in the atmospheric window between 800-1000 cm$^{-1}$ and within the ozone absorption band. AIRS slopes within the atmospheric window are higher than IASI, while they are lower within the ozone absorption band. Since AIRS and IASI differs only in these two spectral regions, the opposite biases could compensate each other when broadband fluxes are calculated giving an almost perfect agreement obtained with broadband lagged regressions (see Figure 1). As for regards the origin for these discrepancies, an important factor that must be considered is the different clear-sky scene selection (see section section 2). The more conservative approach used for AIRS could explain the higher slopes within the atmospheric window. However, these particular aspects will be further discussed in Section 3.4, when the spatial pattern of the LW response is examined.

### 3.3 Atmospheric drivers of the radiative response

The total spectral LW response is now separated into the surface and atmospheric Planck, water vapor and ozone ENSO feedbacks using spectral radiative kernels. Their corresponding flux underwent to the same regression analysis as of IASI and AIRS observations.

The Planck surface feedback (Figure 3, third panel from the top) contributes to the total OLR only within the atmospheric windows, where the OLR is sensitive to the surface. The associated slopes are positive and their maximum is centered at lag equal to +2 months. The Planck atmospheric feedback (Figure 3, fourth panel from the top) causes a positive response throughout all the spectral regions that peaks on average at lag +3 months, except between 645-700 cm$^{-1}$, where the slopes are negative and peak at 5-month lag, explaining the negative slopes observed in the two plots of Figure 2. The positive peak associated to the Planck atmospheric feedback is located in the FIR spectral region, between $\sim$ 400-600 cm$^{-1}$. Here the lag is of +4 months. When the Planck atmospheric feedback is separated in a tropospheric and stratospheric contribution, setting the tropical tropopause layer at 100 hPa, it is clear that the negative slopes come from the stratospheric temperature change (Figure A2) and that the strong positive slopes in the FIR spectral region are from the troposphere (Figure A1), particularly from the upper troposphere (600-150 hPa) (Figure A4). The Planck surface and atmospheric feedback are both negative, enhancing the OLR emission. Water vapor, instead, constitutes the main positive feedback. Indeed, humidity changes cause negative regression slopes throughout the spectrum, meaning that a reduction in emission of OLR takes place. As for the Planck





**Figure 2.** Slopes of the lagged regressions between OLR anomalies and the Niño 3.4 index as a function of wavenumber at a fixed lag of 2 months (top panel), and as a function of wavenumber and lags for IASI (central panel) and AIRS (bottom panel). Shades envelope the 95% confidence interval. Thinner black dots mark 95% significance of the slopes, while thicker black dots mark the lag where the slope peaks at each wavenumber.



atmosphere, the FIR is associated to the highest slopes, with most of the signal coming from the upper troposphere (Figure A4). The water vapor feedback peaks with a lag of +3 months at all wavenumbers where the regression slopes are significant.

Given the strong signal in correspondence of the ozone absorption band, we isolated the ozone contribution as well. A small negative feedback, peaking in the stratosphere (Figure A2), is found around 1025 cm$^{-1}$. It enhances the OLR emission from +2 until +6 months lag. At the same wavenumbers and lags there is also a smaller positive feedback associated to the stratospheric temperature (Figure A2), which, opposed to ozone, acts to reduce the OLR emission.

Finally, we reconstructed the total observed LW response summing the individual feedbacks calculated with kernels. The

feedback sum is shown in Figure 3, second panel from the top, as a function of wavenumber and lag. The individual slopes of water vapor, surface, temperature and ozone feedback, as well as their sum, is shown as a function of wavenumbers for lag equal to +2 months in Figure 3, top panel. The major features of the observed clear-sky OLR response (Figure 2) are present. The kernel regression analysis correctly reproduces the positive slopes within the two atmospheric windows and at the wings of carbon dioxide absorption band, and the negative slopes at the center of it. Their magnitude and time-lag is also

well reproduced. However, there are two notable exceptions: the peak found in correspondence with the ozone absorption band, which is underestimated by the kernel analysis, and the net signal in the FIR and MIR spectral region, which is positive and follows ENSO according to observations, while the opposite is suggested by the kernels sum. A possible explanation could be the fact that ERA5 profiles are referred to all-sky properties, that in the case of water vapor results in an higher humidity content than a clear-sky profile causing an overestimation of the respective feedback, which masks the ozone signal

and overcompensate the atmospheric temperature signal within the FIR and MIR regions. Within the next section the spatial pattern of the OLR response to ENSO is examined, this will help to identify the regions associated to the previous biases.

## 3.4 Spatial pattern of the longwave response to ENSO

We now examine the spatial pattern of the OLR response to ENSO across the tropics (30° S - 30° N), to find the regions associated to the strong ozone response, and to the suspected bias due to the overestimation of the water vapor contribution.

Figure 4 shows the linear regressions between the OLR anomalies in individual spectral channels and the Niño 3.4 index at +2 months lag for every grid box within the considered area. Starting from the top of Figure 4, the maps obtained for IASI (top), AIRS (mid) and the kernel reconstructed signal (bottom) are shown. The left column refers to the channel at 905 cm$^{-1}$, within the atmospheric window, providing information on the surface temperature, but also on the effect of water vapor, that we want to assess and compare among IASI, AIRS and the response reconstructed using spectral radiative kernels.

The right column corresponds to the channel at 1025 cm$^{-1}$, within the ozone absorption band, sensitive mainly to the atmospheric temperature profile and ozone. Here is where the highest positive response within the ozone absorption band - and the whole wavenumber range - is observed. At 905 cm$^{-1}$, a dipole is observed within the tropical Pacific ocean, in phase with the SST pattern typical of El-Niño conditions. Indeed, negative slopes characterize the western part of the basin, while positive values are found in the central and eastern regions. Significant positive contributions are also found over the north-east

subtropical region and over the Maritime Continent. The longwave response at 1025 cm$^{-1}$ is characterized by positive slopes

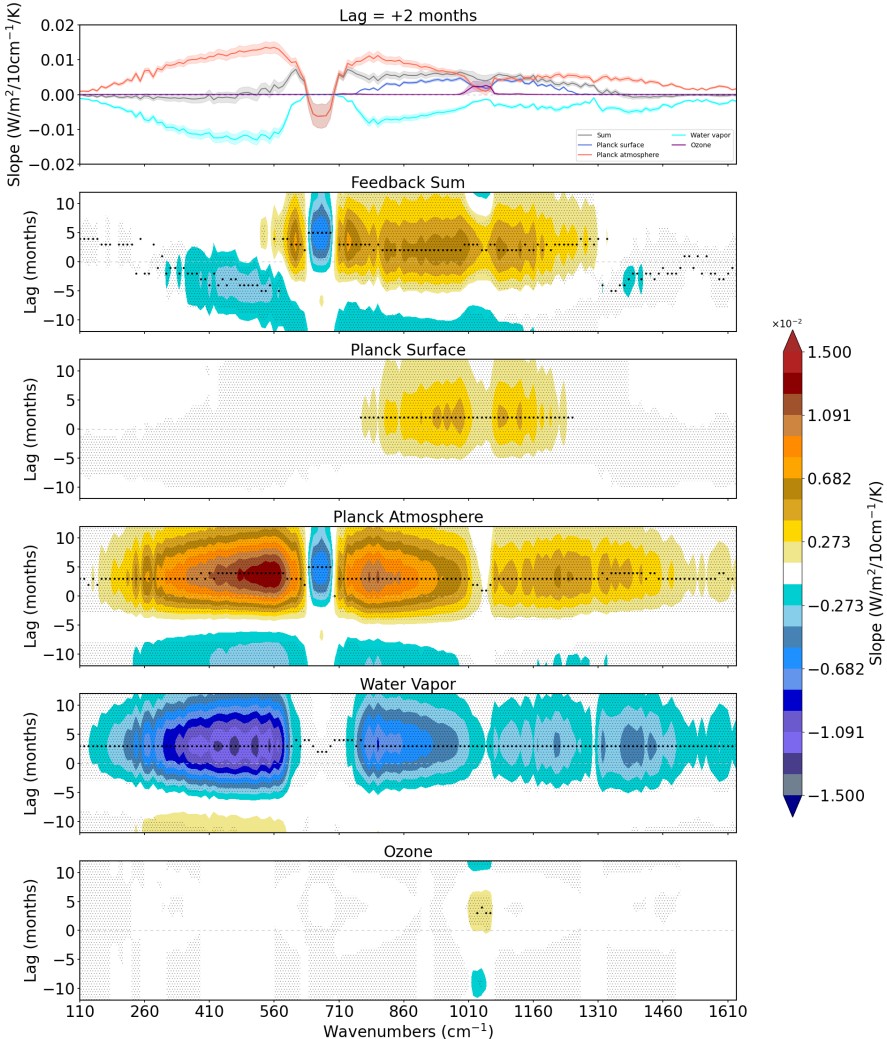

**Figure 3.** Slopes of the lagged regressions between OLR anomalies and the Niño 3.4 index, but for radiative changes decomposed into Planck surface and atmospheric, water vapor and ozone ENSO feedback, and their sum. Slopes are reported as a function of wavenumber at a fixed lag of 2 months in the top panel (feedback sum in gray, Planck atmospheric feedback in red, Planck surface feedback in blue, water vapor feedback in cyan and ozone feedback in purple), and as a function of wavenumber and lags in the panels below. Shades envelope the 95% confidence interval. Thinner black dots mark 95% significance of the slopes, while thicker black dots mark the lag where the slope peaks at each wavenumber.





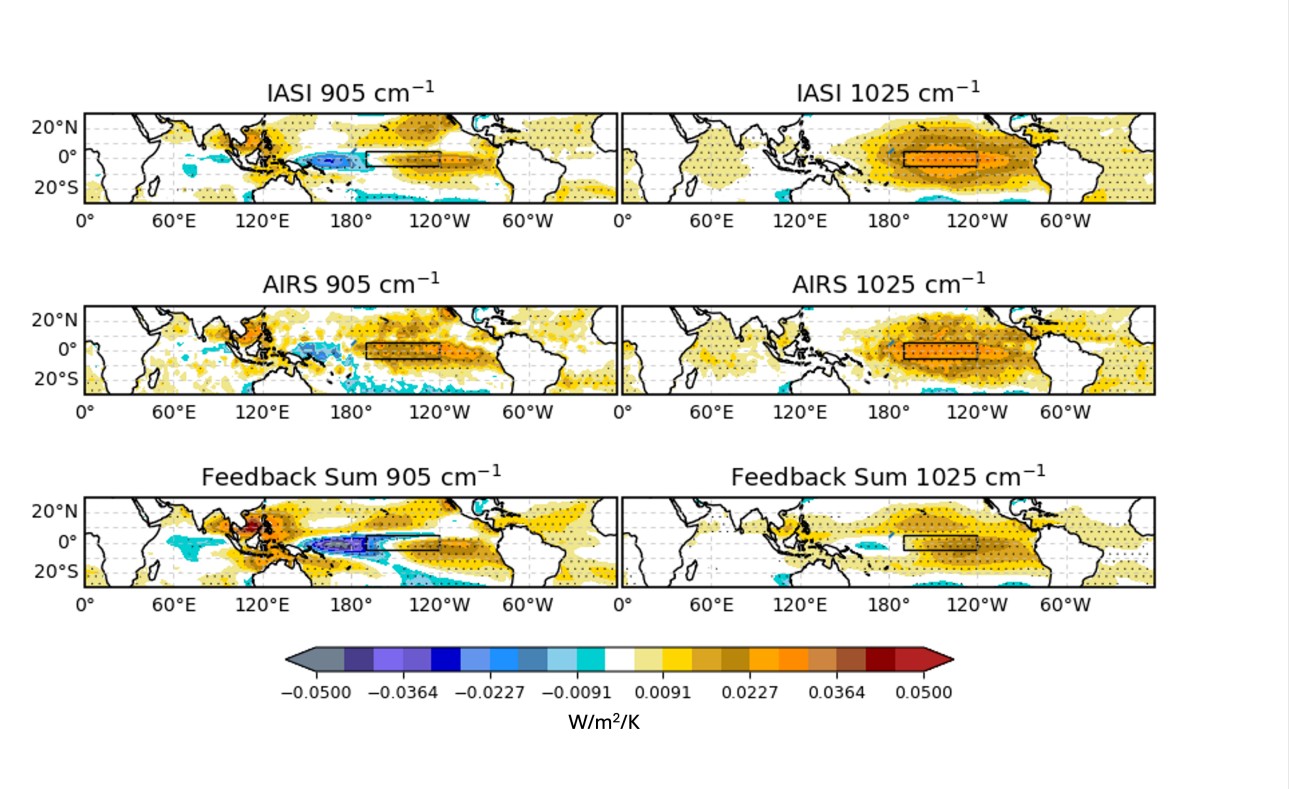

**Figure 4.** Slopes of the linear regressions at lag equal to +2 months between the tropical mean OLR anomalies, at two selected spectral channels, and the Niño 3.4 index, calculated at every grid box within the tropical zone (30°S-30°N). Starting from the top, results for IASI, AIRS OLR and the feedbacks sum. On the left results at 905 $cm^{-1}$, on the right at 1025 $cm^{-1}$. Dots mark regions where the regressions reach the 95% significance level.

coming from the central and eastern Pacific ocean. Their magnitude is highest within the Niño 3.4 region (marked by a black box), and decreases moving off the equator.

Overall, there is a good agreement in the pattern obtained using IASI and AIRS products, although some minor differences can be observed. As already anticipated in Section 3.2, the different temporal and spatial sampling between the two here becomes evident, as highlighted by the more scattered patterns of AIRS with respect to IASI. As mention before, the more conservative cloud filtering of AIRS can explain the slightly positive slopes within the atmospheric windows. In addition to that, the spatial pattern at 905 cm$^{-1}$ shows a less extent region of negative slopes in the west Pacific, constituting another reason for the higher slopes of AIRS. The response pattern is qualitatively well reproduced at 905 cm$^{-1}$ through the kernel reconstruction, although the magnitude of the negative slopes within the west Pacific is clearly overestimated. Higher discrepancies are obtained for the reconstructed response pattern at 1025 cm$^{-1}$. Positive slopes characterize the eastern and central



Pacific but their magnitude is smaller than observations and the peak is shifted eastward, with a region of negative slopes in the western Pacific, partly resembling the 905 cm$^{-1}$ pattern.

## 4 Discussion

The analysis reported in section 3 highlighted some key features of the LW radiative response to ENSO that help to understand
and constrain the main processes behind the variability of the tropical radiative budget.

The main effect of ENSO on the climate system is the anomalous SST pattern across the tropical Pacific Ocean, accompanied by a shift of the Walker Circulation (WC) (Bjerknes, 1966). During the positive El-Niño phase warmer SST anomalies characterize the eastern and central part of the basin, while cooler SST are found in its western part. The WC is shifted eastward, with an increase in convective activity over the central Pacific and a decrease over the West Pacific warm pool. The negative
ENSO phase, La-Niña, brings to the opposite conditions, accompanied by a strengthening of the Walker circulation.

In clear-sky conditions, the climate system responds to ENSO through an increase in OLR emission, that comes primarily from the tropospheric heating caused by the increase in surface temperature and by the increased latent heat release in the upper troposphere. In general agreement with Ceppi and Fueglistaler (2021), this response is observed to reach its peak after and slowly decaying. The observed lag of the radiative response can be explained by the SST pattern effect induced by ENSO,
which causes warm SST anomalies to migrate across the tropical Pacific ocean from the east, at the beginning of an El-Niño event, to the west, where positive anomalies persist up to five months after ENSO peak (Ceppi and Fueglistaler, 2021).

Broadband lagged regressions show a peak of the OLR response at 2-month lag. The spectral analysis shows that channels associated to this lag are the two atmospheric windows, whose radiance is mainly sensitive to the surface temperature. Radiative kernels analysis further support that the delay of the clear-sky OLR response is likely driven by the Planck surface feedback.
Indeed, it is the only one that peaks with a lag of 2 months.

However, a deeper insight is provided by spectrally-resolved OLR measurements. The strongest positive response is associated to the ozone absorption band and peaks at a 3-month lag. Others interesting features are the positive slopes at the wings of the carbon dioxide absorption band, where channels sensitive to tropospheric temperatures, which also peak at 3-month lag, and the negative slopes at the center of it, where radiation is sensitive to the stratospheric temperature. Their peak is the last at
5-month lag. With the aid of spectral radiative kernels, these features can be associated to the relevant physical processes that shape the OLR response to ENSO and its timing.

An increase in the time-lag of the OLR response is observed moving from the lower atmospheric layers to the higher. Specifically, the surface shows the earliest response at a lag of 2 months, followed by the troposphere, where the response peak lags ENSO of 3 months and the stratosphere, where a lag of 5 months is observed. As for regard the magnitude of the response,
it is highest for channels sensitive to the troposphere, followed by channels sensitive to the stratosphere and the surface.

The tropospheric response instead, is shaped by two main feedbacks, the water vapor and the Planck atmosphere. The first informs on the radiative changes driven by humidity changes arising from the anomalous WC during ENSO. An increase in water vapor is related to the ascending branch of the WC. It acts to dampen the emission of OLR (positive feedback), absorbing




most of the radiation emitted by the surface resulting from the increased SST. As a consequence, the main cooling mechanisms

is the emission related to the warming of the troposphere (Huang et al., 2021). The Planck atmosphere feedback is negative enhancing the OLR emission. Therefore, they act to counteract each other and this is true also from a spectral point of view. We see that their lag is the same throughout all the spectrum, except in the FIR between 400 - 600 cm$^{-1}$, where the temperature peaks one month later than water vapor. This evidences that the tropospheric cooling is maximum in this spectral region and is associated to the temperature feedback coming from the upper troposphere.

Moving to the stratosphere, the strong negative feedback within the ozone absorption band, together with the positive feedback associated to the stratospheric temperature, bear evidence of the ENSO stratospheric signal (Zeng and Pyle, 2005; Randel et al., 2009; Calvo et al., 2008; Manzini, 2009; Konopka et al., 2016; Manatsa and Mukwada, 2017; Garfinkel et al., 2018). According to (Domeisen et al., 2019), a reduction in ozone concentration near and above the tropopause is observed as a result of the anomalous upwelling during El-Niño, which cause an increase of ozone-poor air in the central and eastern tropical

Pacific. This agrees with the spatial pattern reported in Figure 4, which show that the ozone feedback is located in regions of increased convective activity. In addition to composition changes, El-Niño also affects the stratospheric temperature. Specifically, an opposite response respect to the troposphere is observed. If the troposphere experiences an increase in temperature during El-Niño, the stratosphere cools. The ENSO impact on stratospheric temperature is the result of the upward propagation of Rossby waves forced by El-Niño and the strengthening of the Brewer Dobson Circulation (BDC) (Domeisen et al., 2019).

The stratospheric temperature decrease related to ENSO can explain the coherent decrease in TOA OLR emission observed at the center of the carbon dioxide absorption band. Despite also involving a stratospheric signal, the OLR decrease is not observed in the ozone band, since the ozone concentration is also co-varying with ENSO near and above the tropopause, reversing the overall signal there. These findings confirm that the ENSO stratospheric signal plays an important role in shaping the LW radiative response peak.

## 295 5 Conclusions

In this work, we used spectrally resolved observation of the OLR to isolate and constrain the main processes controlling the inter-annual variability of the tropical energy budget. We focused on ENSO as the main driver of inter-annual radiative perturbations to the tropical OLR. To the best of our knowledge, this is the first time that the radiative response to ENSO is analyzed using satellite-based spectrally-resolved measurements of the OLR. Results show the potential of spectral OLR to

constrain the fast inter-annual feedbacks and isolate the processes driving them.

We recall here the main results of the study:

- The lagged regression analysis based on the spectrally-integrated tropical OLR from AIRS is largely consistent with CERES, with a slight delay of AIRS, likely due to the spatial sampling strategy used for the radiance to flux conversion. IASI and AIRS agree very well when integrated on the same spectral range, supporting the consistence among different
data products.

- Further insight into the OLR response to ENSO is enabled by the spectral analysis:





- The lag of the response is spectrally dependent, with channels sensitive to lower atmospheric layers peaking earlier than channels sensitive to the stratosphere.

- The peak of the radiative response is found between ∼1000 - 1065 cm$^{-1}$ in correspondence with the ozone absorption band and follows ENSO by 3 months.

- The spectral kernel analysis gives additional information on the processes driving the observed signal:

  - The Planck surface feedback (peaking at 2-months lag) and the Planck atmosphere feedback (3-months lag) are responsible for the negative LW feedback observed throughout the spectrum, that causes an increase in OLR following ENSO. The only positive feedback, which acts to decrease the OLR, is the water vapor feedback, that also peaks at 3-months lag.

  - The response at the center of the carbon dioxide absorption band together with the ozone is likely the result of ENSO induced changes to the stratospheric composition and temperature, which are important for the LW radiative response peak.

  - The sum of the Planck surface, atmosphere, water vapor and ozone feedback generally reproduces the observed signal, though with some differences. In particular, the signal in the ozone absorption band is underestimated, and in the water vapor absorption bands in the FIR and MIR the water vapor feedback perfectly balances the Planck atmosphere feedback, while the observations show that the temperature response prevails. These discrepancies may be caused by an over-estimation of the water vapor response due to how the clear-sky fields of ERA5 are obtained.

In the FIR spectral region, both the water vapor and temperature ENSO feedbacks are higher, but to date we could only rely on simulated spectral fluxes at these wavenumbers. In this regard, an important contribution will be provided by the future Far-infrared Outgoing Understanding and Monitoring (FORUM) mission, the selected Earth Explore-9 (EE-9) of the European Space Agency (ESA) (Palchetti et al., 2020), that will close this observational gap, allowing to assess the role of this critical region for the inter-annual and forced climate feedbacks.

This study sets the basis for a diagnostic of ENSO-induced variability of the OLR, based on spectrally resolved measurements, that will allow to better disentangle the various processes contributing to ENSO OLR response, avoiding compensation errors that may arise by using the broadband OLR. We plan to apply this diagnostic to evaluate climate models participating to the Coupled Model Inter-comparison Project phase 6 (CMIP6) in a upcoming work, which could improve our insight on how models reproduce the coupled atmosphere-ocean dynamics driving the ENSO response.

*Data availability.* The CERES EBAF OLR fluxes are made available by NASA Langley Atmospheric Science Data Center (NASA/LARC/S-D/ASDC), Distributed Active Archive Center (DAAC) via https://doi.org/10.5067/TERRA-AQUA/CERES/EBAF-TOA_L3B004.1 (NASA/LARC/SD/ASDC, 2019). AIRS Level 3 spectral fluxes are provided by Goddard Earth Sciences Data and Information Services Center (GES DISC) via https://doi.org/10.5067/5P7KQ31XI7XJ (Huang, 2020). IASI Level 3 spectral fluxes are provided by the Free University of Brussels/Laboratory for the study of Atmospheres, Environments, and Space Observations (ULB/LTMOS) via https://iasi-ft.eu/data-access/OLR/)



(Whitburn, 2021a). ERA5 reanalysis products can be downloaded from Copernicus Climate Change Service (C3S) Climate Data Store (CDS)

via https://doi.org/10.24381/cds.6860a573 (Hersbach et al., 2023a)

.

## Appendix A

**Figure A1.** Slopes of the lagged regressions between OLR anomalies and the Niño 3.4 index, but for tropospheric radiative changes decomposed into atmospheric, water vapor and ozone ENSO feedback. The tropical tropopause is set at 100 $hPa$. Thinner black dots mark 95% significance of the slopes, while thicker black dots mark the lag where the slope peaks at each wavenumber.







**Figure A2.** Same as Figure A1, but for radiative changes within the stratosphere.





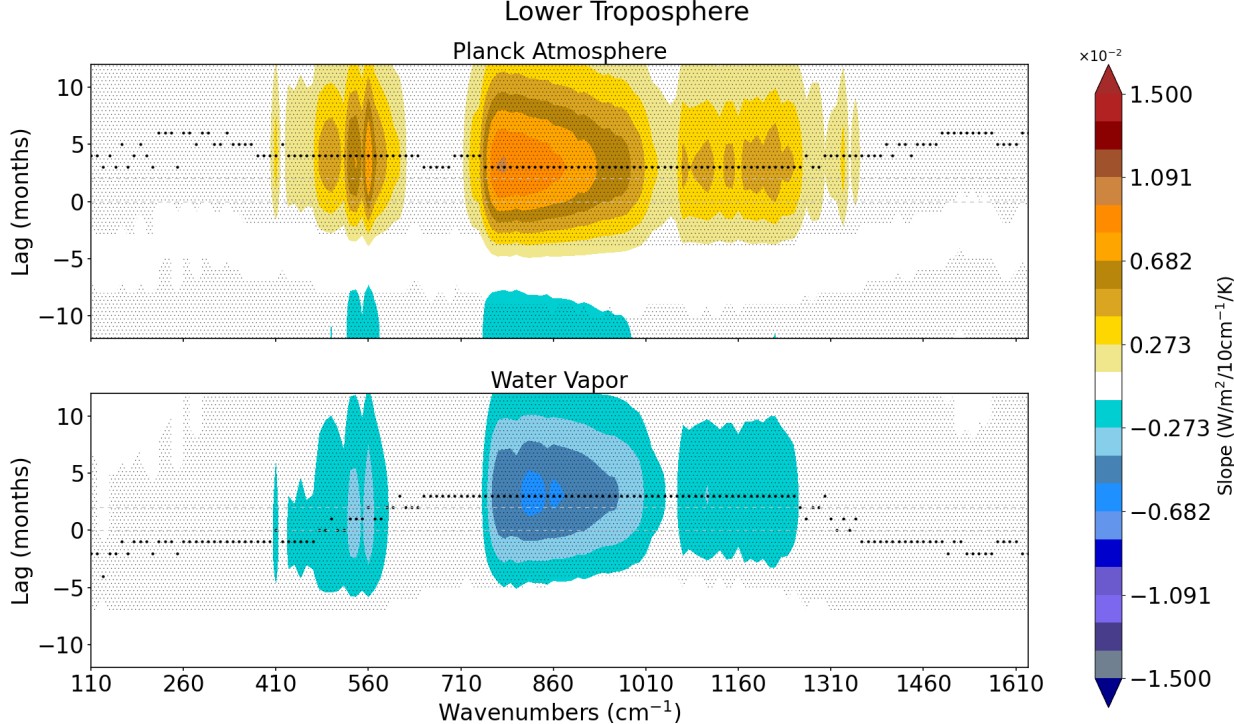

**Figure A3.** Same as Figure A1, but for radiative changes within the lower troposphere (1000-600 $hPa$).





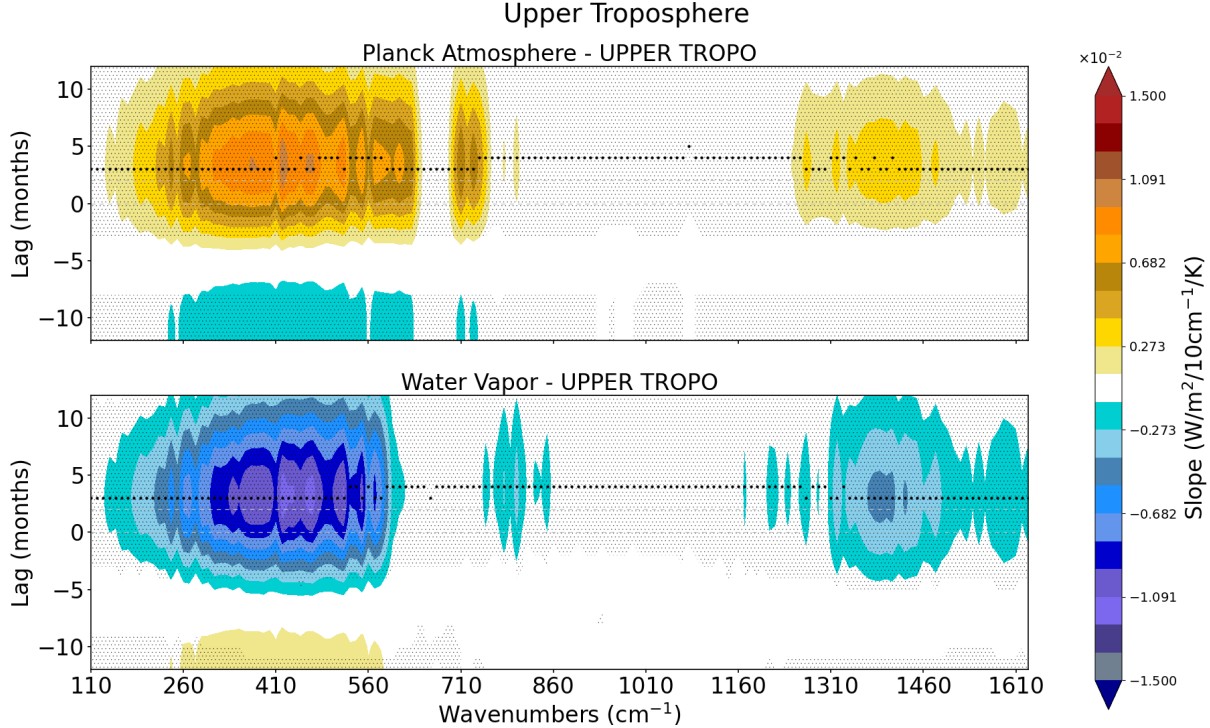

**Figure A4.** Same as Figure A1, but for radiative changes within the upper troposphere (600-150 $hPa$).

*Author contributions.* "B.M. Dinelli and E. Castelli supervised the work. F. Fabiano and S. Della Fera designed the experiments and M. Taddia carried out the formal analysis and wrote the manuscript with contributions from all co-authors."

*Competing interests.* All the authors declare that no competing interests are present.

*Acknowledgements.* This work has been supported by SERCO Italia s.p.a, who funded part of the PhD fellowship of M. Taddia, within which this study has been developed. The authors also thanks F. M. Palmeiro for the productive discussion on the stratospheric dynamics and Professor X. Huang for the useful consultation which helped the set up of the beginning stages of this work.



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
