# Peer review of "A Spectral Perspective of ENSO Driven OLR Variability"

_EGUsphere, 2025_

## Author Comment (AC3)

**Reviewer 1:**

This is a very interesting work that extends the diagnoses of the important OLR-Ts relationship in ENSO from broadband to spectral radiation. The development and application of a spectral kernel dataset is a notable strength (despite some questions detailed below). The paper shows that spectral data makes it possible to view the lagged OLR-TS relation with more clues relatable to the geophysical variables at the process level, which makes an important finding. I would recommend publication if the following comments were addressed.

- The Introduction gives a good literature review to motivate this work, with a rather complete collection of relevant previous works, despite some misses, for example, Huang and Ramaswamy 2008 (https://doi.org/10.1029/2008GL034859) which was one of the earliest data-based diagnosis of spectral OLR-Ts relationship.

We thank the reviewer to highlight this reference that we missed. It contributes to set the basis of the work and will be included in the introduction to expand the present literature review.

- On the other hand, I found the paper does not make as strong connections with the previous works when discussing the results, which potentially impedes the revelation of the novelty or difference in the findings here. For example, much emphasis of the paper is on the lagged OLR-Ts relation, for which the spectral signatures are related to geophysical drivers (Figs 2/3 and texts around Line 200). However, these findings were similarly made based on broadband kernel-decompositions (e.g., Fig 3 of Kolly & Huang). Given the objective of this paper is to demonstrate the advantage of spectral information, such comparisons can be used to discuss what additional information is brought in by the spectral data.

This touches on a very critical point of the analysis, specifically, its ultimate goal and we want to be as straightforward as possible in communicating this. The discussion of the results, as emerged from the review, needs to be improved to make this point clearer and stronger. With respect to previous studies of ENSO-driven radiative feedbacks that used broadband longwave (LW) fluxes (for example Kolly and Huang (2018)), we would like to emphasize two main outcomes. The use of the spectral dimension allowed us to investigate the spectral (wavenumber) dependency of the OLR response and the vertical propagation of ENSO signal. This information is directly contained in the measurement, without the need of decomposing it through the kernel analysis. The decomposition is performed to study the origin of the signal in the different spectral regions. In addition, as also performed in Kolly and Huang (2018), this additional information can be used to evaluate climate models performances. These points will be discussed in more depth and with more attention in relation to the existing literature when preparing the revised manuscript.

- A main, technical comment is on the kernels themselves, which are computed based on monthly profiles (Line 120) – a simplification known to introduce biases (e.g., see Huang and Ramaswamy 2009, https://doi.org/10.1175/2009JCLI2874.1). It remains to be demonstrated how well the kernels produced here explain the total

spectral OLR changes. Although it is qualitatively discussed (Line 210), the biases are not quantified – better to show the residuals in closure tests of different lags. Another suggestion is to compare the kernels produced here to other kernels. For example, the broadband radiative sensitivity values spectrally integrated from the spectral kernels should reproduce broadband ERA5 kernels truthfully computed from instantaneous profiles (Huang and Huang 2023, https://doi.org/10.5194/essd-15-3001-2023). Such comparisons can be made with respect to maps of vertically integrated kernel values and with kernel-reproduced lagged OLR-Ts relationship (Fig 1). This would help provide a measure of the uncertainty in the results.

We agree with the reviewer that a closure test would be needed to quantify the biases between the observed OLR response and the one reconstructed using the kernels method.

We are aware of the shortcomings that arise from the use of monthly rather than instantaneous profiles to calculate radiative kernels. Although this simplification mostly impacts the accuracy of kernels when the all-sky scenario is considered, we performed a comparison between our kernels and the ones computed by Huang and Huang 2023 to confirm this preliminary consideration. Two tests have been made: radiative kernels used in this work have been compared against kernels provided by Huang and Huang 2023; then we computed the spectral kernels with input profiles at a higher temporal resolution. Specifically, in the new calculation, we sampled 20 time steps (corresponding to 5 days at 6-hour intervals) during one month and we averaged the derivatives computed at each time step. As an example, we show a comparison of the surface temperature kernels computed for a single month with the corresponding kernel computed by Huang and Huang (2023) integrated on a similar spectral range, Huang and Huang (2023) from 10 to 2600 $cm^{-1}$ and this work from 105 to 2755 $cm^{-1}$. In the left map of Figure 1, the percentage differences are shown using the new computation, while on the right we show the results using the monthly average surface temperature.

[Figure]

*Figure 1. Differences between clear-sky surface temperature radiative kernels computed using hourly (new) and monthly (old) data with respect to radiative kernels computed by Huang and Huang (2023).*

Even though this is only an example limited to a single variable, the differences are small and do not appear to be significant for the results of our analysis. Nevertheless, we will explore this aspect in more detail and discuss it further in the new version of the manuscript.

- Given the results are exclusively for clear-sky. It is better to indicate this in the paper title.

We agree with the reviewer and will modify the title accordingly. A new title could be: "An analysis of ENSO driven OLR variability in clear-sky conditions from a spectral point of view" or "A spectral perspective of the clear-sky OLR variability driven by ENSO"

- Lastly, there are numerous grammatical errors. The English writing needs to be thoroughly proofed/edited.

We thank the reviewer for the accurate reading. The revised manuscript will undergo a substantial revision/rewriting with respect to English grammar and terminology.

**Reviewer 2:**

The authors investigate the radiative response to ENSO from a spectrally resolved perspective. The underlying idea is that spectrally resolved OLR holds information about individual contributions to radiative changes from temperature, humidity, and ozone, which cannot be resolved by broad-band OLR. The authors compute spectral kernels combining ERA5 data with a radiative transfer model. Comparing the spectral kernels with the satellite observations allows to identify changes in surface temperature, air temperature, humidity, and ozone, and to further decompose these changes into contributions from different atmospheric levels. The authors find the changes one would expect from ENSO in the spectrally resolved OLR product.

I read this paper with great interest and recommend publication of the study under the condition that the comments below are addressed. I divide my comments roughly into major, minor, and suggestions that I don't require to be addressed. I have some expertise with radiative feedbacks and tropical dynamics, but almost no experience with satellite observations.

"major"

- The authors missed the opportunity to strictly decompose the observed spectral OLR into contributions from the individual feedback processes using spectral fingerprints, as was done for example in Huang et al. 2010 (https://doi.org/10.1029/2009JD012766). While I would be very interested to see this, it would be exaggerated to ask the authors to perform this analysis, and the paper contains valuable insights even without this. However, I still mention this here, because it implies some limitations to the interpretations that the authors give.

We thank the reviewer for the suggestion. Indeed, Huang et al. (2010) used an optimal detection method to calculate the spectral fingerprint of the climate change induced by a doubling of the $CO_2$ concentration. They use the partial radiative perturbation (PRP) method as a reference to test optimal detection results. While in this work we calculate a spectral fingerprint by using spectral radiative kernels – an alternative method to PRP to separate individual contributions of radiative feedbacks – the application of a similar optimal detection method, as the one described in in Huang et al., (2010), would allow us to directly calculate radiative feedback from IASI and AIRS observed spectra without the use of radiative kernels. This is an interesting opportunity that can be explored in the future.

- l. 9 – 10: This is the first example where it matters. I strongly suggest to reword this sentence in the abstract because there was no strict mathematical "decomposition". Rather, the satellite observations were compared by eye to the kernels to identify the feedback contributions.

*L. 9-10: "The observed signal is then decomposed using a spectral kernel analysis into water vapor, surface and air temperature, and ozone feedback, to evaluate the role of individual processes building the overall response."*

We agree that the term decomposition might be misleading in this context. We will modify this throughout the manuscript. This can be rewritten as: "The spectral fingerprint of water vapor, surface and air temperature, and ozone feedback is then calculated using a set of spectral kernels to evaluate the role of individual processes building the overall response."

- Huang et al. 2010 should be cited in the introduction as an example for how spectral OLR can be used to identify individual feedbacks

This article will be mentioned to expand the current discussion with a description of the different methods that can be used to study radiative feedbacks. As previously mentioned, the main advantage of this study is that it enables the spectral fingerprint of radiative feedback to be estimated directly from the observed data. Thank you for bringing this to our attention.

- l. 69, l. 329: I don't quite understand what is meant here by "diagnostic". If it refers to the decomposition into feedbacks using spectrally resolved OLR, then this should not be referred to as new, because it has already been done by Huang et al.

In the context of lines 69 and 329, we would like to emphasize the future implications of this study, which is the comparison of the ENSO spectral fingerprint obtained from data such as IASI and AIRS spectral fluxes, to the same quantity simulated by general circulation models (GCMs). The term "diagnostic" refers to the spectral fingerprint of ENSO obtained from IASI and AIRS observations, which can be compared to that reconstructed from climate model outputs. As the original sentence was too generic, the revised manuscript will expand on this aspect, in a similar way to what has been done here.

- l. 115 – 122: It is not clear to me how the kernels are computed from the data. In particular, I would like to know how the derivatives were computed. A (finite) partial derivative needs radiation computations from at least two different atmospheric states, and I don't understand from the description which these are. I assume that in each partial derivative computation only one variable was perturbed, but how / to what value?

The finite partial derivatives have been calculated using the RTTOV model, for which this calculation is implemented. More details on the computation of the spectral kernels used for this work are now provided by Della Fera et al., (2025).

- l. 155: I found no further investigation of this, can you point me to where this is done?

We thank the reviewer for pointing this error out. Indeed, this is mentioned in Section 3.4, Lines 235–236, and not in Section 4 as stated. The correct reference will be inserted in the revised manuscript. However, we would also like to take this opportunity to highlight the implications of the different clear-sky scene selection of AIRS and IASI in the Discussion or Conclusions section.

- All figures: "slope" is not a very descriptive y-label. l. 110 – 112 state that this slope is referred to as "ENSO feedback", so maybe "feedback" in Fig. 1, or "spectral feedback" in Fig. 2-4 could be an option?

We agree that "feedback" or "spectral feedback" is a more descriptive label that could improve the readability of figures showing the regression coefficients of lagged regressions. We take the suggestion made by the reviewer and the term "ENSO feedback" will be used as the y-label in Figure 1, while "spectral feedback" will be used in Figures 2–4.

- Figs. 2, 3, 4: The color map should have linearly increasing lightness and brightness. The current color bar artificially suggests strong gradients where there are none. For example, with increasing values the colors get darker until 0.455 in Fig. 2 and then brighter again, creating false perceptions of the actual values. Some hues even seem to repeat. I find https://doi.org/10.1038/s41467-020-19160-7 to be a very helpful resource for picking a color map.

We thank the reviewer for suggesting a resource for contour plot color bars and for highlighting this important aspect of presenting our results. In the revised manuscript a proper color bar with linearly increasing colors will be used to improve the readability of figures showing a gradient.

- Figs. 2 and 3 should use the same value range so that they can be accurately compared to each other

As well as a more proper color bar, the same value range will also be used to enable a better comparison between different plots.

- l. 182: The fact that there is a transition from analysing satellite observations to analysing something derived from the spectral kernels and reanalysis should be made clear.

This passage will be highlighted at the beginning of Section 3.3 "Atmospheric drivers of the radiative response".

- Section 3.4, Fig. 4, also l. 284-286: How sure can we be that this signal comes from ozone? While this spectral band surely represents ozone where ozone is optically thick, the positive values in the Eastern Pacific could also originate from increased surface emissions through the atmospheric window if ozone is not optically thick there. A fingerprint analysis might have been able to clearly separate between ozone emissions and surface emissions. Given that this has not been done, can we be sure how much of this is ozone signal and how much is surface signal? My doubt is intensified by the fact that the 905 cm^-1 channel which purely represents surface emissions also shows increases in the Eastern Pacific.

This is an important question to be addressed for discussing the results of our analysis. We provide additional results below that may help with the discussion. Figure 2 shows the feedback across the Tropical Pacific at 905 cm$^{-1}$ (left panel) and 1025 cm$^{-1}$ (right panel). We want to recall that spectral kernels (as well as IASI and AIRS spectral fluxes) provide the flux integrated over 10 cm$^{-1}$ bands (W/m$^2$/10 cm$^{-1}$). Therefore, the two wavenumbers selected are not single spectral channels.

[Figure]

*Figure 2. Spatial pattern of ENSO feedbacks at 905* cm⁻¹ *(left panel) and 1025* cm⁻¹ *(right panel) at the 2-months lag. From top to bottom Planck surface, Planck atmosphere, water vapor, ozone feedback and their sum. Thin black dots mark regression coefficients at the 95% confidence level.*

At 1025 cm⁻¹ the signal of ozone and Planck surface feedback are simultaneously present in the central and eastern tropical Pacific. While their magnitude is almost the same, their spatial pattern differs. The signal related to surface temperature is limited to the equator, while the ozone signal extends off to the north and south of it. Instead, at 905 cm⁻¹ the Planck surface feedback is higher in magnitude than at 1025 cm⁻¹ and the ozone feedback is almost null. Based on the kernel analysis, it can be concluded that at 1025 cm⁻¹ both the Planck surface and ozone feedback contribute to the TOA OLR response to ENSO, but the ozone feedback dominates.

These maps are not included in the current version of the manuscript, but they will be added to the appendix of the revised version and the discussion will be expanded accordingly.

- l. 272: The humidity changes from ENSO are complicated and may also arise from the generally warmed atmosphere. Without further analysis or citations it is not totally clear to me that the positive water vapor feedback comes from the ascending branch of the Walker circulation. In particular in the dry regions, radiation is also extremely sensitive to small humidity changes. Furthermore, even though this study treats clear-sky regions, cloud (un)masking can play a role for apparent water vapor feedback.

We thank the reviewer for the insight. We agree with the reviewer that the conclusion drawn from line 272 is oversimplifying, we will rephrase and extend the discussion on this point in the revised manuscript to consider other possible explanations.

- l. 277-279: Where does this statement come from? Why tropospheric cooling? Doesn't an increased OLR in the FIR imply a warmed troposphere? The sentence as a whole is confusing to me. Cooling cannot maximize in a spectral region, only radiation can.

This statement is incorrect as far as it is formulated and misleading, we therefore thank the reviewer for commenting on that. The sentence can be rephrase: "The maximum emission of OLR via atmospheric temperature feedback occurs one month later than that of water vapor between 400 and 600 cm$^{-1}$. This suggests that this spectral region plays an important role in the cooling of the system following ENSO."

- Major edits are needed with respect to writing and grammar

We thank the reviewer for the accurate reading. The revised manuscript will undergo a substantial revision/rewriting with respect to English grammar and terminology.

- I suggest to indicate that clear-sky OLR is studied (and not simply OLR) in the title

We agree with the reviewer that a change in the title is necessary, given the implications of the use of clear-sky observations (e.g. different cloud masking). As already anticipated within the answers to the first review, a new title could be: "An analysis of ENSO driven OLR variability in clear-sky conditions from a spectral point of view" or "A spectral perspective of the clear-sky OLR variability driven by ENSO".

"**minor**"

- l. 48: reproduced by models, I assume?

Line 48 states: *"Indeed, there are evidences that the radiative response to ENSO is still not completely reproduced in both its amplitude and timing (Kolly and Huang, 2018; Planton et al., 2021; Ceppi and Fueglistaler, 2021)."*

We thank the reviewer for pointing this out. The sentence refers to climate models performances. It can be rewritten as: "Indeed, there are evidences that the radiative response to ENSO is still not completely reproduced in both its amplitude and timing by climate models (Kolly and Huang, 2018; Planton et al., 2021; Ceppi and Fueglistaler, 2021)."

- l. 77: most of the paper is written in terms of wavenumber, but here wavelength is used. I suggest to stick to one, preferably wavenumbers

We agree with the reviewer that using the same term throughout the text would be preferable. The same term "wavenumber" will be consistently used throughout the revised manuscript.

- l. 123 – 134: This is all fine but was hard for me to understand. It could be due to my lacking expertise of satellites. For example, I only understood in the Discussion section that FORUM is a satellite mission, and even a quick google didn't help me understand what the "synthetic" in "synthetic radiance" refers to (although I could figure it out eventually). This paragraph could be improved with small edits.

The term "synthetic" in "synthetic radiance" indicates that the radiance has been simulated. This comment will be taken into account when the paragraph is improved, as the reviewer suggested.

- l. 161: "maps": they are not maps

"**suggestions"**

- I don't require these to be addressed.
- l. 195: This is really counter-intuitive, because a positive value corresponds to a negative feedback. I see how this arises from the fact that OLR is defined positive outward, which makes sense. Still, can this confusion be overcome? Maybe by showing -dOLR/dT in the figures, or by referring to net downward LW instead of OLR, or by explicitly stating that a positive value implies a negative feedback?

We agree with the reviewer that the opposite sign of the results presented by the contour plots makes it difficult their comprehension, as well as their discussion. Therefore, we plan to show -dOLR/dT in the figures of the revised manuscript.

- Fig. 2 first panel would profit from a line at y=0
- l. 329 – 333: Without finger printing, it is not clear to me how this method can be used for a quantitative analysis to evaluate climate models. Furthermore, when comparing Figs. 2 and 3 it seems that ERA5 does a good job, except in the FIR where the satellite anyway doesn't really know what's going on, because it's extrapolated. So how would this add anything beyond directly comparing the feedback components due to surface temperature, air temperature, water vapor, etc. between ERA5 and climate models? These are directly available from the output.

We appreciate this comment because it gives us feedback on how clearly we explained the main objective of our analysis. Secondly, it enables us to improve our explanation. In this regard, we want to emphasize that the actual diagnostic is that provided by IASI and AIRS observations. The kernels analysis is used to gain a deeper insight in the information provided by spectral satellite observations and to evaluate how well we can reconstruct the spectral response with the kernel technique. In addition, as the reviewer points out, the calculation of radiative perturbations driven by changes in specific climate variables can be performed using the simulated variable fields and radiative kernels of climate models, as we did with ERA5.

- l. 319 – 323: it could be made clearer that "The sum of the Planck surface, atmosphere, water vapor and ozone feedback" refers to ERA5 and the "observed signal" to the satellite.
- In my opinion, the spectral kernels are the coolest contribution of this paper to the scientific community. It would be amazing to have them publicly available. :)

We are glad to see that the reviewer is interested in this data product. A reference for the radiative kernels used in this study is now publicly available in Della Fera et al., (2025). This reference will also be included in the revised manuscript.

**Reviewer 3**

I read this manuscript with great interest. The study addresses the radiative response to ENSO from a spectrally resolved perspective, which provides valuable insights into the contributions of individual processes. By combining CERES broadband observations with spectrally resolved measurements from AIRS and IASI, and by employing radiative kernels derived from ERA5 reanalysis, the authors disentangle the roles of temperature, water vapor, and ozone in shaping the clear-sky OLR variability associated with ENSO. The spectral dimension is a particularly valuable addition, as it helps separate overlapping processes that cannot be distinguished in broadband analyses.

I found the introduction to be well written and comprehensive, setting a solid foundation for the work. However, the remainder of the manuscript often lacks rigor, and the quality of the English requires substantial revision.

My comments are structured in three categories: General, major, and minor, as detailed below. I would recommend publication of this manuscript provided that the authors carefully address the issues raised in this review.

General comment:

- While the study offers interesting insights into the spectral perspective of ENSO-driven OLR variability, I found that several aspects of the analysis lack the level of rigor and quantitative support.

Thank you for your careful reading and highlighting these important issues that we had overlooked.

- First, the comparisons presented in the manuscript are mostly qualitative. For example, differences between instruments or between kernels and observations are described verbally but rarely quantified. Providing numerical values (e.g., absolute differences, percentage differences) or plots of differences (e.g. for Fig. 2 and Fig. 4) would significantly strengthen the robustness of the analysis.

The first objective of our analysis was to explore OLR variability driven by ENSO from a spectral point of view. As the reviewer suggested, taking a more quantitative approach — for example, quantifying the differences between the observational datasets used, and between the observations and the reanalysis results — would lead to stronger conclusions. Therefore, this aspect will be carefully checked before the submission of the revised manuscript.

- Second, unless I overlooked it, the authors do not discuss the impact of the difference in overpass time between AIRS and IASI as a potential explanation for the discrepancies observed in the atmospheric window regions. Even when restricted to ocean-only observations, such differences have already been reported in the literature (e.g., Whitburn et al., 2020), and one likely contributing factor is indeed the distinct overpass time of the two instruments. This issue should be explicitly acknowledged and discussed.

We thank the reviewer for bringing this to our attention. Since the analysis results obtained by IASI and AIRS show differences within the atmospheric window (between 800

and 1000 cm⁻¹), the impact of the different overpass time of the two instruments can be discussed as a potential cause for that discrepancy. As shown by Whitburn et al., (2020) this should have a small impact on the spectral fluxes over the ocean, however, as suggested by the reviewer this issue should be acknowledged. This will be added within the "Data and Methods" section, when the datasets employed for the analysis are described.

- Third, it is important to note that Metop-A started drifting from its orbit in June 2017. The manuscript does not discuss the potential impact of this drift on the results. Considering Metop-B data from 2018 onwards could help avoid potential biases. Addressing this point would increase confidence in the robustness of the conclusions.

The orbital drift of Metop-A could impact the overpass time. This in turn would affect radiative fluxes within the atmospheric windows, which are the spectral regions that are more sensitive to the surface temperature. In addition to the different overpass times of IASI and AIRS, the orbital drift of the Metop-A will be mentioned as a potential cause of the bias observed between the two data products. However, we believe that the discrepancy between the IASI and AIRS data products is most likely the result of different cloud masks applied.

Major comments:

- Lines 104-105: The manuscript mentions "[…] removing […] the linear trend for the whole period." Could the authors provide more details on the method used to remove the linear trend? In particular, it would be important to confirm and explicitly emphasize in the manuscript that this step is intended to remove the effect of increasing greenhouse gas concentrations over the study period.
  At the same time, the authors should discuss the implications of this methodological choice. Assuming a uniform linear trend may not fully capture the real evolution of the climate system, which can be non-linear (e.g., acceleration of warming in recent decades). There is also a risk that part of the signal interacting with ENSO could be inadvertently removed. While detrending is a reasonable approach to isolate interannual variability, the rationale and limitations of this choice should be clearly stated.

The linear trend (LT) of the OLR anomaly has been calculated in the following way. A linear regression has been performed between the OLR anomaly time series and the monthly dates:

$$OLR\ anomaly\ TS\ =\ m \cdot monthly\ dates\ +\ q$$

Where, *OLR anomaly TS* is the time series (TS) of the OLR anomaly from 2008 to 2020 and *monthly dates* are the years from 2008 to 2020 considered by the analysis. The regression coefficients $m$ and $q$ obtained have been used to calculate the linear trend (LT):

$$LT\ =\ m \cdot monthly\ dates\ +\ q$$

the LT reconstructed has then been subtracted to the OLR anomaly TS:

$$OLR\ anomaly\ TS\ detrended\ =\ OLR\ anomaly\ TS\ -\ LT$$

To discuss the implications and limitations for the analysis, the lagged regressions of IASI and AIRS data will be performed for both the OLR anomaly TS with and without the LT removed.

- From my understanding of the manuscript, the study treats El Niño and La Niña phases equivalently by relying on a linear regression with the Niño 3.4 index to evaluate the radiative impact. However, El Niño and La Niña are not necessarily mirror images of each other: their amplitudes, spatial structures, and temporal lags can differ significantly. Could the authors clarify whether an analysis was performed separately for El Niño years and La Niña years? If so, were the results consistent? If not, at least a discussion of this potential limitation would be important to assess the robustness of the conclusions.

We thank the reviewer for addressing this additional point of discussion. We confirm that no separate analysis of El-Niño/La-Niña events has been performed. In this regard, we acknowledge that this work investigates the overall ENSO activity, thus the results obtained are indicative of the overall behavior of this phenomenon. However, we thank the reviewer for this suggestion. A separate analysis of El Niño and La Niña events may offer new insights, and this will be considered in future developments of the study.

- Lines 127-128: The spectral flux is computed using the Gaussian quadrature method with only three angles. I wonder whether this choice provides sufficient accuracy for the purposes of the study. I suggest that the authors evaluate the potential impact of using a larger number of angles (e.g., 4 or 5) on at least one or two representative cases, to demonstrate that the results are not sensitive to this assumption.

Thank you for this comment. In our work, the spectral radiative kernels are calculated starting from simulated radiances at three different angles and a Gaussian quadrature is then applied for an approximate computation of the radiative flux. As suggested, to address the impact of this method for the results of the analysis, we conducted a sensitivity test on the number of angles used for the radiance-to-flux conversion. Due to the computational time required, we performed a test on kernels computed for a single month (January 2008) using both 3 angles (as in the manuscript) and 4 angles. The sets of angles used were taken from Clough et al. (1992). As an example, Figure 3 presents the percentage differences between the surface temperature kernel computed using 4 and 3 view angles. As can be seen, these differences are limited to within 1% and are not significant. However, this may not be the case for the all-sky scenario.

[Figure]

*Figure 3. Percentage difference between the radiative kernel (W/m²) of the surface temperature (Ts) for the month of January 2008 calculated using 4 and 3 angles.*

- The manuscript primarily discusses the slopes of the lagged regressions, but the correlation coefficient (R) or the fraction of variance explained is almost never reported. Including R or $R^2$ would be important to assess how much of the variability in spectral fluxes is actually explained by ENSO. Reporting these values would strengthen the robustness and interpretability of the results.

As anticipated, we agree that improving the strength of the final statements through more accurate statistical considerations would benefit the analysis robustness. In this regard, we will therefore include a plot of the correlation coefficient and/or the $R^2$.

- The slopes of the lagged regressions are expressed in $W\,m^{-2}\,10\,cm^{-1}\,K^{-1}$. While this is certainly relevant to evaluate the radiative response to ENSO, the analysis would benefit from also discussing the changes in relative terms, for example as a percentage or in terms of Brightness Temperature. Indeed, regions where the slope is largest often correspond to wavenumbers with the highest radiance, rather than where the ENSO response itself is maximized. Including such normalized metrics would provide a clearer and more physically meaningful interpretation of the spectral response.

We agree that a more objective insight would be provided by expressing the regression coefficients in relative terms. We will include a discussion of the results in terms of percentage change in the revised manuscript.

- Lines 175-176: "Since AIRS and IASI differ only in these two spectral regions, the opposite biases could compensate each other [...]". Could the authors clarify whether this compensation actually occurs when the total (integrated) flux is calculated?

The compensation would occur when IASI and AIRS spectral fluxes are integrated over the same wavenumber range. This statement is based on the fact that the IASI and AIRS broadband fluxes calculated over the 645–1995 cm$^{-1}$ wavenumber range demonstrate an

excellent level of agreement. The peak of the radiative response shows the same magnitude and 2-months lag. When the regression coefficient spectra of IASI and AIRS are plotted at a 2-month lag (see Figure 2, upper panel), differences emerge in the atmospheric window between 750 and 1000 cm$^{-1}$, as well as within the ozone absorption band around 1043 cm$^{-1}$. Specifically, IASI exhibits a lower response within the atmospheric window, and a higher response within the ozone absorption band, with respect to AIRS. Given that their agreement is almost perfect when the broadband OLR is considered and that the IASI and AIRS signals are in good agreement at other wavenumbers, it was hypothesised that these biases would cancel each other out. We will consider removing this sentence since it does not have any implication for the rest of the work.

- The manuscript does not seem to discuss the uncertainty associated with the reconstructed radiances in the spectral ranges not covered by AIRS. Could the authors provide an evaluation or at least an estimate of these uncertainties? Such information is crucial to assess the robustness and reliability of the results in those spectral regions.

We agree that it would be important to discuss the uncertainty associated with AIRS reconstructed fluxes. We will do our best to address this issue in the final manuscript.

- Lines 223-224: To evaluate the spatial pattern of the OLR response to ENSO, the authors select a channel at 905 cm$^{-1}$ to extract information on surface temperature and the effect of water vapor. However, no significant $H_2O$ absorption line is present at 905 cm$^{-1}$. Would it not be more appropriate to consider the OLR integrated over a spectral range within the window region? This would better capture the combined effects of surface temperature and any residual water vapor absorption.

The authors recognized that there might have been a misunderstanding. As the sentence is currently formulated, it is unclear whether the spectral radiances are expressed as W/m$^2$/10 cm$^{-1}$. This would mean that they are integrated over bins of 10 cm$^{-1}$. As such, 905 cm$^{-1}$ is not a single wavenumber.

- Lines 234–235: The manuscript states, "As already anticipated in section 3.2, the different temporal and spatial sampling between the two here becomes evident, as highlighted by the more scattered patterns of AIRS with respect to IASI." However, in lines 100–101, it is mentioned that all datasets have been regridded to a common 2.5° × 2.5° grid using bilinear interpolation, and both datasets correspond to monthly mean fluxes. It is therefore unclear why AIRS would appear more scattered than IASI. Could the authors clarify the origin of this apparent discrepancy? For instance, is it due to residual effects from cloud screening, incomplete coverage, or some other factor?

We agree that this sentence is not properly formulated, and that the words "temporal sampling" and "spatial sampling" are not correctly used in this context. Indeed, as the reviewer points out the two datasets have the same monthly temporal resolution and have been interpolated to the same horizontal grid. The main reason behind the more scattered pattern of AIRS is therefore suggested to be the cloud screening applied to the

two datasets. Figure 3 reports the percentage of OLR values at each grid box over the period from January 2008 to December 2021 for the two overpass times in AIRS (upper row) and IASI (lower row) datasets. The maps show that the amount of valid points follows the cloud cover over the tropical regions. Another factor that may play a role is the strategy employed to derive AIRS fluxes, which is based on a collocation strategy with respect to CERES footprint (Huang et al., (2008)). The "different spatial sampling" would refer to this last, maybe these words are misleading and we will change them in the revised manuscript.

[Figure]

*Figure 3. Number of valid points (percentage %) for each overpass times of the OLR time series from January 2008 to December 2020 in AIRS (upper row) and IASI (lower row) datasets.*

- The Discussion section needs significant rewriting. Several passages are unclear or imprecise, and the analysis would benefit from more quantitative support. References to figures are often missing, making it difficult for the reader to link statements to the presented data; the authors should clearly indicate which figure panels support each point. Additionally, the last paragraph should include a reference to Whitburn et al., 2021, as similar points were discussed there.

On the basis of all the comments emerged during this first review phase, the section Discussion will surely undergo rewriting. It will benefit from a quantitative evaluation of the results, as well as stronger connections to other results presented within existing literature on related subjects.

Minor comments:

- Lines 18-19: The sentence "It contains both the signature of the increased concentration of radiatively active species [...]" is slightly misleading. The spectral OLR contains the signatures of all radiatively active species, not only those whose

concentrations increase. The current formulation is correct when referring specifically to OLR trends.

Maybe this can be rewritten as: "It contains both the signature of climate variables, such as radiatively active species [...]"

- Lines 87-88**:** The statement "AIRS fluxes are calculated exploiting the same angular distribution models of CERES [...]" is not fully clear. CERES provides broadband OLR, while AIRS produces spectrally resolved fluxes at 10 cm$^{-1}$. Could the authors clarify how the CERES ADMs are applied to AIRS data in this context?

We thank the reviewer for the careful reading. This sentence is the result of misinterpretation. The same scene type information from the CERES footprint and auxiliary information has been used for AIRS (so the ADMs are not the same as stated). This approach is based on a collocation strategy. Only data from AIRS pixels that overlap with the CERES footprint up to a certain threshold are retained for the radiance-to-flux conversion. In addition to the cloud filtering, this is another reason for the more scattered pattern of AIRS.

- Line 92: [...] "from 15 to 1995 cm$^{-1}$". Isn't it from 10 to 2000 cm$^{-1}$?

The dataset used for this study, that has been downloaded from https://disc.gsfc.nasa.gov/datasets/AIRSIL3MSOLR_6.1/summary, provides spectral fluxes from 15 to 1995 cm$^{-1}$ (every 10 cm$^{-1}$).

- Line 95: Metop-C was launched in 2017 and not in 2013.

The authors thank the reviewer for highlighting this error. The correct year will be indicated in the revised version of the manuscript.

- It could be useful to include a plot of the Niño 3.4 index for the study period, so that readers can better visualize the ENSO variability. In addition, a brief discussion of the relevant ENSO phases during this period would provide valuable context for the analysis.

We agree that this would contribute positively to the comprehension of the analysis. Inserting a plot of the Niño 3.4 index could help the discussion of the feedback sign and spatial pattern. As shown by Figure 4 the overall OLR response follows the SSTs anomalies of El-Niño phase: an increase in TOA OLR emission is found across the central and eastern Tropical Pacific, while a decrease is suggested to characterize the western part of the basin.

- Lines 154–155: The manuscript mentions that the difference is "likely due to AIRS's clear-sky scene selection, as will be investigated further in section 4." However, I could not find a further discussion of this point in section 4. Could the authors clarify or correct this statement?

We thank the reviewer for pointing this error out. Indeed, this is mentioned in Section 3.4, Lines 235–236, and not in Section 4 as stated. The correct reference will be inserted in the revised manuscript. However, we would also like to take this opportunity to highlight the implications of the different clear-sky scene selection of AIRS and IASI in the Discussion or Conclusions section.

● Lines 196–197: For clarity, it would be helpful to remind the reader that a negative slope corresponds to a decrease in the TOA OLR, which implies that more radiation is trapped and therefore corresponds to a positive feedback (and conversely).

These lines could be rewritten as: "The positive regression slopes that characterize the Planck and atmospheric feedback indicate an increase in OLR emission at the TOA. Consequently, they constitute a negative feedback that acts to stabilize the system. Conversely, changes in humidity cause negative regression slopes throughout the spectrum, meaning that a reduction in OLR emission takes place at the TOA, constituting positive feedback." However, we will also take into consideration to plot -dOLR/dTs.

● Figure 2: Why does the x-axis stop at 1515 cm$^{-1}$, whereas Figure 3 extends further? For consistency, it would be preferable to use the same spectral range in both figures unless there is a specific reason for this choice.

As suggested also by Review 2, the figures in the revised manuscript will be improved by changing the colour map and y-label, and by using the same magnitude scale in all plots.

● Lines 205–215: This paragraph is not very clear and would benefit from some rewriting. In particular, the statement *"the net signal in the FIR and MIR spectral region"* is too vague. Could the authors specify the exact wavenumber ranges referred to and indicate clearly in which panel(s) of the corresponding figure this signal can be seen?

We agree that generally referring to the "MIR" and "FIR" is too vague and the wavenumbers of the actual spectral regions will be specified in the revised manuscript. As suggested the paragraph will also undergo a careful revision to make the description of the results as clearer as possible.

● Figure 3: For clarity, it might be better to move panels 1 and 2 to the bottom of the figure, as they are discussed later in the text. This would make it easier for the reader to follow the discussion in the correct order.

● A study of ENSO teleconnections beyond 30° N/S could be an interesting addition to the conclusion or perspectives. Extending the analysis to the extratropics might provide further insights into the global radiative response to ENSO.

The authors thank the reviewer for the suggestion, which adds value to the analysis. We will take it into consideration to extend the future perspective of the study during the writing of the revised manuscript.

● In the Conclusion, the authors state: "to the best of our knowledge, this is the first time that the radiative response to ENSO is analyzed using satellite-based spectrally-resolved measurements of the OLR." This statement is not entirely accurate, as similar analyses, albeit to a lesser extent, were conducted in Whitburn et al., 2021.

We thank the reviewer for highlighting that we are missing some relevant reference that contributes to setting the context for this analysis. A comment on Whitburn et al., (2020) will be added within the Discussion and Conclusions sections.